# Metabolic Alteration of *Trypanosoma cruzi* during Differentiation of Epimastigote to Trypomastigote Forms

**DOI:** 10.3390/pathogens11020268

**Published:** 2022-02-19

**Authors:** Salvatore G. De-Simone, Saulo C. Bourguignon, Priscila S. Gonçalves, Guilherme C. Lechuga, David W. Provance

**Affiliations:** 1Center for Technological Development in Health (CDTS), FIOCRUZ, National Institute of Science and Technology for Innovation in Neglected Populations Diseases (INCT-IDPN), Rio de Janeiro 21040-900, RJ, Brazil; priscilla.desimone@cdts.fiocruz.br (P.S.G.); guilherme.lechuga@cdts.fiocruz.br (G.C.L.); bill.provance@cdts.fiocruz.br (D.W.P.J.); 2Epidemiology and Molecular Systematic Laboratory, Oswaldo Cruz Institute, FIOCRUZ, Rio de Janeiro 21040-900, RJ, Brazil; 3Cellular and Molecular Biology Department, Biology Institute, Federal Fluminense University, Niterói 24020-141, RJ, Brazil; saulocb@id.uff.br

**Keywords:** *Trypanosoma cruzi*, carbohydrate metabolism, organic acid, high-performance liquid chromatography, carboxylic acid, enzyme activity

## Abstract

Intracellular parasites such as *Trypanosoma cruzi* need to acquire valuable carbon sources from the host cell to replicate. Here, we investigated the energetic metabolism of *T. cruzi* during metacyclogenesis through the determination of enzymatic activities and quantification by HPLC of glycolytic and Krebs cycle short-chain carboxylic acids. Altered concentrations in pyruvate, acetate, succinate, and glycerate were measured during the growth of epimastigote in the complex medium BHI and their differentiation to trypomastigotes in the chemically defined medium, TAU3AAG. These alterations should represent significant differential metabolic modifications utilized by either form to generate energy. This paper is the first work dealing with the intracellular organic acid concentration measurement in *T. cruzi* parasites. Although it confirms the previous assumption of the importance of carbohydrate metabolism, it yields an essential improvement in *T. cruzi* metabolism knowledge.

## 1. Introduction

*Trypanosoma cruzi*, the etiological agent of Chagas disease, presents during its life cycle three primary morphologically and physiologically distinct evolutive forms designated as epimastigote, trypomastigote, and amastigote [1]. As the epimastigote (epi) forms are non-infective, a critical step in the lifecycle of *T. cruzi* is its transformation into the highly infective metacyclic trypomastigote (trypo) forms. This process, designated as metacyclogenesis, can be reproduced in vitro using several specific growth conditions [2,3,4], including a chemically defined medium [5,6]. Besides the transformations of epimastigotes, other ways of metacyclogenesis also occur in the vector [7,8].

It is known that the metacyclogenesis of these parasites leads to some morphological [9,10] and biochemical changes that include alterations in the surface carbohydrates [11], variations in the lipid composition [12,13], and fatty acids [14]. Previous studies have reported that both the epimastigote and trypomastigote forms can obtain energy from glucose, amino acids, fatty acids, and glycerol metabolism [15,16,17,18,19,20]. Yet, despite the various metabolic studies on *T. cruzi*, the principal substrate for energy generation during differentiation is a controversial topic. Organic acids have been suggested as the primary substrate for energy generation in trypomastigote [15,21]. On the other hand, glucose, rather than amino acids, has been indicated as the principal substrate in epimastigotes. It has also been postulated that this parasite oxides amino acids to generate ATP [22,23].

Although the basic knowledge about trypanosomatid carbohydrate metabolism and its particularities has been unraveled in the previous decades [24,25], new information has become available in recent years, showing that the process is more elaborate and has additional information on peculiar features [13,26,27,28]. These features have been evaluated through enzymatic measurements [16,29,30], enzyme expression studies [16,28,29,30], predictions through computational approaches [31], gene expression/genomic analyses [32,33,34], and metabolomics [26,35].

The use of high-performance liquid chromatography (HPLC) to analyze organic acids in biomedical and pharmacological applications [36,37,38] has opened the possibility of investigating their concentrations as the final products of diverse enzymatic activities under different cellular growth conditions. In the present work, we focused our attention on determining the short-chain carboxylic acids produced from glycolysis and Krebs cycle intermediates by ion-exchange high-performance liquid chromatography during the metacyclogenesis of *T. cruzi* parasites. Furthermore, these organic acid concentrations were compared to parasites grown under different culturing conditions.

## 2. Results

### 2.1. Organic Acids

Separate organic acid analyses were performed in populations of parasites grown in a brain heath infusion (BHI) complex and TAU3AAG chemically defined media. Samples of parasites cultured in BHI medium were harvested in the logarithmic phase at 24, 72, and 120 h and in the stationary phase at 168 h to follow the organic acid profile during the transition. The organic acid profile was monitored during the differentiation process using samples withdrawn at 24, 48, and 72 h of culture in TAU3AAG medium. At 24 h, the parasite population consisted of about 75% epimastigotes and 25% trypomastigotes forms. Differentiation reached nearly 50% at 48 h, and by 72 h, the parasite population consisted mainly of the trypomastigote form (70%). The harvested parasite samples were processed for high-performance liquid chromatography (HPLC) analysis, as described in the materials and methods section, to identify the organic acids extracted by comparing retention times to standards. The organic acid concentrations were obtained by integrating the chromatographic peak area with means. Figure 1 graphically represents the concentration variations of these acids extracted from BHI parasites, and Figure 2 for those extracted from TAU3AAG parasites (Appendix A).

Appendix A show the concentrations of parasite-derived organic acids cultured in BHI and TAU3AAG medium, respectively.

The G2P concentration obtained from parasites cultured in BHI media was maintained high in the various time points studied, with a slight decrease from 120 to 168 h. The citrate, acetate, and malate presented a similar profile: a significant increase in concentration from 24 to 120 h, with a fall at 168 h, adequate only for acetate and succinate. Pyruvate, malate, and succinate concentrations remained low, but a significant increase in pyruvate concentration was observed from 120 to 168 h. On the other hand, malate concentration decreased after 72 h and remained stable at the lowest levels (Figure 1).

During metacyclogenesis in TAU3AAG media, the intermediate acids involved in the glycolytic pathway (glycerate-2-phosphate (G2P) and pyruvate) decreased, as well as the acetate that presented a one- to twofold decrease in concentration (Figure 2). Despite this decrease, however, the attention of the G2P remained high compared to the other acids. Succinate was the only acid that increased concentration (four- to fivefold) from 24 to 48 h, but the whole time at low concentrations along with the culture (Figure 2; Appendix A).

### 2.2. Enzymatic Activities

At different differentiation times under TAU3AAG or BHI media, samples of parasites were collected and extracted as described to evaluate the enzymatic activity of the aldolase, pyruvate kinase, and hexokinase. The results are shown in Figure 3 and Figure 4 and Appendix A. Pk and Ald activity decreased significantly after 24 h and after 48 h for Hk. After 72 h, a small but significant increase in Pk and Hk activity was observed.

In the total extract from epimastigotes cultured in BHI media (Figure 4), the aldolase activity remained constant and appeared enhanced in the lag and log phases (24–72 h) as well as at the beginning of the stationary phase (120 h). On the other hand, the enzymatic activity of pyruvate kinase and hexokinase at 120 h of culture in BHI media increased four- to fivefold and threefold from 24 h, respectively. Comparing the profiles of enzyme activity in the extracts of parasites maintained in TAU3AAG (Figure 3), and cultured in BHI media (Figure 4), the activity of three enzymes (PK, HK, and Ald) in the glycolytic pathway dropped during the first 24 h of differentiation. However, there is no significant difference (Appendix A) in the pyruvate kinase activity in epimastigotes between 24 h, 72 h, and 120 h.

## 3. Discussion

Several studies have been performed on the energetic metabolism in trypanosomatids by evaluating enzymatic activities or through computational approaches. Yet, few have quantified the intermediate metabolic products of the different enzymatic reactions as a means to investigate differences between organisms or their stages of development. Here, an HPLC analysis of short-chain carboxylic acid from glycolytic and Krebs cycle of *T. cruzi* showed differences in the concentration of various compounds during its growth in BHI medium (Figure 1) and its differentiation in TAU3AAG medium (Figure 2). These differences indicate that both energetic pathways are modulated and most likely reflect the dramatic changes in the physiological state of the parasite [26,39,40].

At the end of the differentiation process (72 h in TAU3AAG medium), the high amounts of G2P and the low level of pyruvic acid (Figure 2) suggest that these values could be the consequence of the allosteric regulation of the glycolytic pathway. In trypanosomatids, the glycolytic pathway is not regulated similarly to the classical control points of hexokinase and phosphofructokinase of bacteria and mammalian cells. Instead, pyruvate kinase is the subject of solid allosteric regulation [15,24,39]. The low enzymatic activity of pyruvate kinase (Figure 3) and the pyruvate concentration agree with this regulation point. Therefore, if the glycolytic pathway operates at a basal level at any given moment, it can be supposed that it utilizes another path(s) to generate ATP.

One hypothesis is that during differentiation in the TAU3AAG medium, proteins from reservosomes organelles [41] can be used to generate ATP. However, Figueiredo et al. [42] showed that the endogenous energy reserve, present in epimastigote and trypomastigote forms, is not a glycogen-like polysaccharide but is represented by lipids and protein, respectively. Still, during the aging of a *T. cruzi* culture, type I reservosomes contain an electron-dense matrix with lipids, and type II reservosomes present a homogeneous matrix without lipid inclusion but with the presence of proteins [19,43,44]. Furthermore, these organelles were absent in highly infective bloodstream trypomastigote forms isolated from infected mice [43]. Therefore, these results indicated a significant difference in metabolic reserves within these different evolutive forms of trypanosomatids [41,45,46].

The decrease in enzymatic activity of aldolase, pyruvate kinase, and hexokinase (Figure 3 and Figure 4) that associated with the reduction in the levels of G2P and pyruvate (Figure 2), found in our study indicates that during metacyclogenesis or morphological change of the parasite, the glycolytic pathway or glucose consumption is reduced.

These results follow the data described by Adroher et al. 1990 [16]. These authors compared the enzymatic activity of three enzymes (hexokinase, phosphofructokinase, and pyruvate kinase) in epimastigotes and metacyclic of *T. cruzi.* Higher activity of the two first enzymes was found in epimastigotes and metacyclic, whereas pyruvate kinase had similar activity in both forms of the parasite. The specific activity is an important parameter and demonstrated clearly that in this work, a quantitative change in the glycolytic pathway of both culture forms of *T. cruzi*. Furthermore, a similar observation with the decreasing glycolytic enzymatic activity was obtained during amastigote differentiation to epimastigote forms [16,45]. 

Likewise, Barisón et al. [34] used metabolomics to access the levels of 47 metabolites both in the exponential and stationary phases of *T. cruzi* epimastigotes observed that there was also a reduction in the levels of citrate, pyruvate, malate when parasites in the stationary phase were compared to parasites in the exponential phase.

Therefore, it appears that this metabolic change is a general process that precedes the differentiation processes of amastigote to epimastigote, epimastigote to trypomastigote, and possibly trypomastigote to amastigote. However, this presumably does not reflect the natural changes in the rectum of the vector, the leading site of metacyclogenesis, since the presence of glucose is questionable [46].

Conversely, little is known about the possibilities of using externally available and internally stored fatty acids as resources to survive in nutrient-poor environments and to sustain metacyclogenesis. For example, studies conducted with a parasite population consisting of 99% in the epimastigote form showed that the conversion of palmitic acid into CO2 was minimum, which indicated that the use of this fatty acid is irrelevant in this life stage [21]. However, the same authors showed that the conversion of palmitic acid into CO2 increased in proportion to the appearance of the trypomastigote form, thus suggesting the usage of lipid reserves [21,47].

The endogenous energy reserve present in various evolutive forms of T. cruzi has been attributed to carbohydrates [19]. However, while epimastigotes can interchange glucose and amino acids, they also can accumulate fatty acids into lipid droplets during growth and use this reserve to support its growth after glucose exhaustion [13,48,49]. Furthermore, this same study demonstrated that the insect stages coordinate the activation of fatty acid consumption with the metabolism of glucose [48].

Maintaining a high enzymatic activity in the glycolytic pathway during the culture of epimastigote forms in the BHI medium (Figure 4) indicates that the primary energy source in this form should be carbohydrates. These findings confirm a previous result [20], although the low activity of glycolytic enzymes and intermediates, along with differentiation, suggests that glycolysis is not the primary pathway used during metacyclogenesis. This followed another study conducted that measured the metabolites produced and excreted during culture conditions with proton nuclear magnetic resonance spectroscopy [20]. The final products of glucose catabolism differed quantitatively and qualitatively for the three lifecycle stages of *T. cruzi*. To a lesser extent, the end products of metabolism produced by epimastigote forms were mainly acetate and pyruvate and L-alanine and ethanol. Differences between epimastigotes and metacyclic forms were only quantitative. However, accessible amastigotes and amastigote-like forms excreted acetate, glycerol, pyruvate, and to lesser extent succinate, but no L-alanine or ethanol [18].

Thus, the association of both results with the disappearance of the lipids in the reservosomes [26] suggests whether or not epimastigotes use monosaccharides in the vector and oxidize through beta-oxidation before the end of metacyclogenesis. These organelles are absent in amastigotes and trypomastigotes, being exclusive to epimastigotes of *T. cruzi* [42,43,44]. However, despite this strong evidence, additional studies are needed to demonstrate that beta-oxidation is activated in the differentiation process or if it participates together with the oxidation of the amino acids [13,24].

The oscillations in the concentration of the intermediates from the Krebs cycle (malate and succinate) demonstrate that the metabolic alterations during differentiation can also affect the main oxidative pathway in *T. cruzi* [50,51,52,53].

## 4. Materials and Methods

### 4.1. Parasites and Growth Condition 

The Dm28c clone of *Trypanosoma cruzi* was maintained as previously described [17] and harvested epimastigote forms after growth in BHI medium for 1, 3, 5, and 7 days. For metacyclogenesis, epimastigotes harvested by centrifugation from a five-day culture in BHI were washed and incubated at room temperature for 2 h in a solution of triatomine artificial urine (TAU; 190 mM NaCl, 17 mM KCl, 2 mM MgCl_2_, 2 mM CaCl_2_, 0.035% sodium bicarbonate, 8 mM phosphate, pH 6.8) and three days at 28 °C in TAU3AAG (TAU containing 10 mM l-proline, 50 mM sodium l-glutamate, 2 mM sodium l-aspartate, and 10 mM glucose) at a density of 5 × 10^6^ cells/mL.

### 4.2. Extraction of Organic Acids

Parasites were washed twice by centrifugation in PBS before loading into a screw-capped culture tube fitted with a Teflon-line cap. An extraction reagent (200 μL of 18 N H_2_SO_4_, 0.6 g of NaCl, 5 mL of diethyl ether, and 25 μL of acetonitrile) was added and blended on a vortex mixer for 1 min before centrifugation (1000× *g* for 1 min). The ether phase added 0.2 mL of 0.1 N NaOH, and the tubes were gently shaken. The lower NaOH phase was tested with a pH paper, and 0.1 N NaOH was added, one drop at a time, to obtain a pH of 9.0. Next, the tubes were vortexed for 1 min, centrifuged as before, and removed and discarded the ether phase. Twenty-five microliters of acetonitrile were added, and the tubes were left uncapped for about 5 min to allow the residual ether to evaporate. 

### 4.3. HPLC Analysis

Samples of 80 μL were injected into a cation exchange HPLC column (Aminex HPX-87H, 300 × 7.8 mm, ID, Bio-Rad Laboratories Inc. São Paulo, Brazil) previously equilibrated with 0.007 N H_2_SO_4_. The organic acids were fractionated for 60 min at 35 °C on an automatic HPLC system (6A model; Schimadzu, Kyoto, Japan) at a flow rate of 0.6 mL min^−1^ using a full-scale unit absorbance of 0.08 wavelength of 210 nm. The column was calibrated in the same solution with a standard organic acid mixture (citrate, tartrate, malate, succinate, and acetate) purchased from Bio-Rad Laboratories Inc. (São Paulo, Brazil) to determine the relative concentration. Other organic acids (glycerate-2-phosphate and pyruvate) were dissolved in 0.07 N H2O4 to prepare solution stocks. Quantity was determined using standards of organic acid concentration to calibrate the data module for the units processed by the system integrator. Values of the analysis correspond to the mean ± SD of three biological replicates.

### 4.4. Enzymatic Assay and Protein Estimation

Hexokinase and pyruvate kinase enzymes were assayed at 25 °C in a final volume of 1 mL, and changes were measured in absorbance at 340 nm for 5 min and measured and expressed as specific activities (µmol_·_min^−1^_·_mg^−1^ of protein) as described [16]. In addition, aldolase activity was measured by a colorimetric assay kit (MAK223, Sigma-Merck, St. Louis, MO, USA) and protein concentration using a colorimetric assay based on the Bradford method (Bio-Rad Laboratories Inc., São Paulo, Brazil).

### 4.5. Statistical Analysis

Statistical analyses were performed on GraphPad Prism version 5.03 (GraphPad Software, San Diego, CA, USA) using a one-way ANOVA test and Bonferroni post-test to compare results, and data were considered statistically different when *p* ≤ 0.05.

## 5. Conclusions

We performed a qualitative and quantitative analysis of organic acids during the growth and differentiation of *T. cruzi*. The variation in oxaloacetate detection performed in this study compared to other studies using a different Trypanosomatidae was ≥ 2% [17]. Comparative studies of oxaloacetate and enzymatic activity confirmed that glucose is the primary energy source of epimastigote forms in a glucose medium. The Krebs cycle was functional in the oxidative succinate-oxaloacetate direction. A decrease in the utilization of the glycolytic pathway occurred, despite the high concentration of glucose in the extracellular medium during the differentiation of epimastigote to trypomastigote forms in TAU3AAG medium. This fact was confirmed by the reduction in glycerate-2-phosphate and pyruvate concentration and the decrease in the activity of the enzymes hexokinase, aldolase, and pyruvate kinase. These findings suggest that other energy factors such as lipids and proteins can be used during differentiation.

## Figures and Tables

**Figure 1 pathogens-11-00268-f001:**
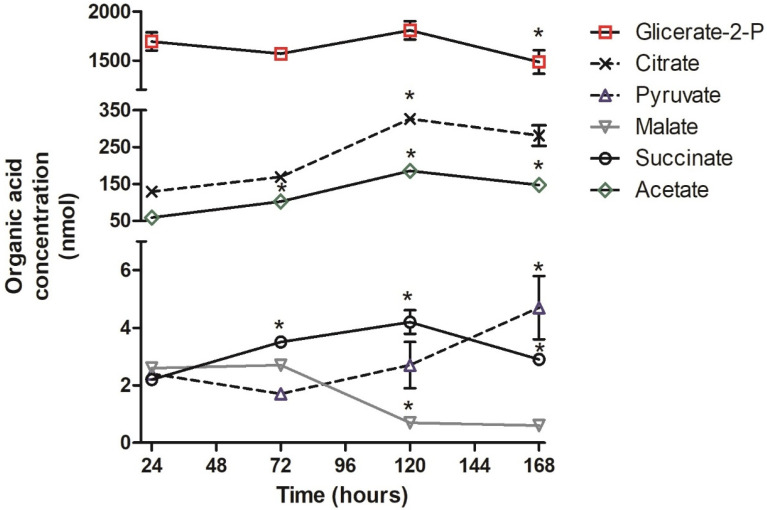
Concentration of parasites’ organic acids after different growth periods as a function of growth time in BHI media. The concentrations of succinate (−O−), malate (−∇−), and pyruvate (−Δ−) were nearly equivalent and appeared superimposed, while citrate (−X−) and acetate (−◊−) show similar profiles during cultivation. However, the glycerate 2*p* (−□−) concentration was noticeably higher than the others were. Values correspond to the mean ± SD (three biological replicates) of organic acid concentration measured in three independent cultures from a sample size of 3 × 10^8^ parasites. *: *p* ≤ 0.05, statistically significant from previous time point.

**Figure 2 pathogens-11-00268-f002:**
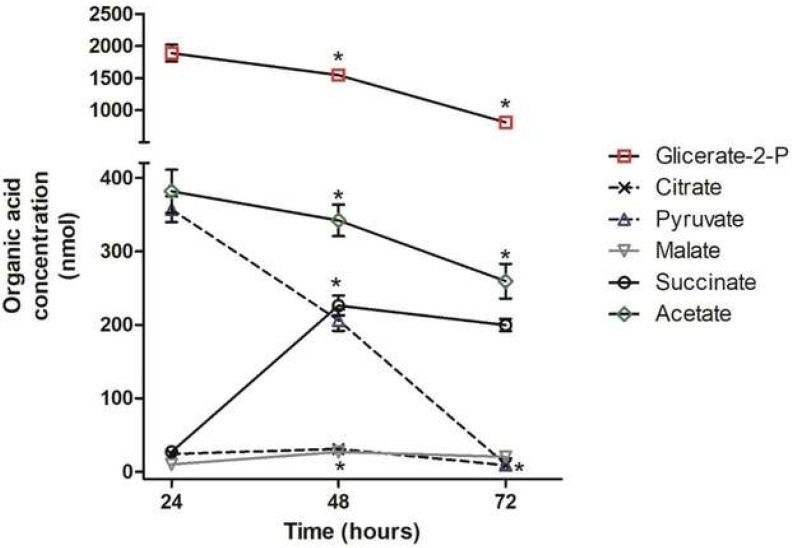
Concentration of the organic acid from parasites in different times of the cultivation in TAU3AAG medium: glycerate 2*p* (−□−), citrate (−X−), pyruvate (−Δ−), malate (−∇−), acetate (−◊−) and succinate (−O−). The values correspond to the mean ± S.D (three biological replicates) of the concentration in nmoles of organic acid of three experiments determined in a population of 3 × 10^8^ parasites. *: *p* ≤ 0.05, statistically significant from previous time point.

**Figure 3 pathogens-11-00268-f003:**
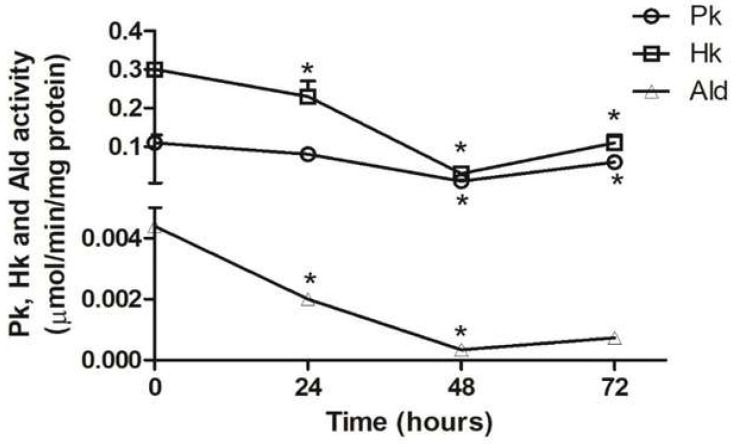
Specific enzymatic activity of pyruvate kinase (Pk), hexokinase (Hk), and aldolase (Ald) in *T. cruzi* during its differentiation to trypomastigotes in TAU3AAG medium. The percentage of trypomastigotes forms were < 1%, 25%, 50%, and 70% at 0, 24, 48, and 72 h, respectively. *: *p* ≤ 0.05, statistically significant from previous time point.

**Figure 4 pathogens-11-00268-f004:**
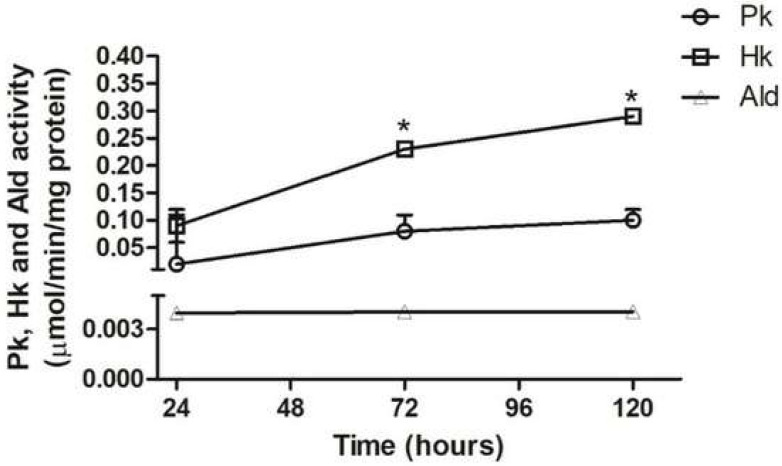
Activity of pyruvate kinase (Pk), hexokinase (Hk), and aldolase (Ald) in *T. cruzi* cultured in BHI medium for different times. Extracts from 2 × 10^8^ parasites were assayed in three independent experiments, and the values plotted correspond to the average. The unit for Pk and Hk was µmoles subst/min/mg protein, and for the aldolase nmoles subst/min/mg protein and means an S.D. are presented. *: *p* ≤ 0.05, statistically significant from previous time point.

## Data Availability

The data presented in this study are available on request from the corresponding author.

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
