# Peer review of "Metabolic Alteration of Trypanosoma cruzi during Differentiation of Epimastigote to Trypomastigote Forms"

_pathogens, 2022, doi:10.3390/pathogens11020268_

Round 1

Reviewer 1 Report

In the work entitled ‘Metabolic alteration of Trypanosoma cruzi during differentiation of epimastigote to trypomastigote forms’ De-Simone and co-authors use HPLC to quantify glycolytic and Krebs cycles short-chain carboxylic acids as well as to measure the enzymatic activities of aldolase, pyruvate kinase, and hexokinase in different time points of the life cycle of T. cruzi, the causing agent of Chagas disease. The authors found alterations in the levels of pyruvate, acetate, succinate, and glycerate when epimastigotes were compared to metacyclic trypomastigotes. Additionally, the authors show a strong reduction in the enzymatic activities of aldolase, pyruvate kinase, and hexokinase during the differentiation of epimastigote to the metacyclic trypomastigote form. Overall, the data is well presented. However, parts of the text need extensive English language and style editing to improve the clarity of the work. Moreover, Introduction, Material and Methods, Results and Discussion sections must be improved.

Introduction

Page 1, Lines 31 /32:

“…presents during its life cycle three main/major morphologically and physiologically distinct evolutive forms…”

Although the authors do not mention stages such as the intracellular epimastigotes (Almeida-de-Faria et al, 1999), adding the word ‘major’ or ‘main’ will implicate that there are other life stages of T. cruzi beyond the three mentioned stages.

Page 1, Lines 34 / 35

“…is its transformation into the highly infective metacyclic trypomastigote (trypo) forms.”

Although the authors mention the main morphological stages, it is important to differentiate metacyclic trypomastigotes from bloodstream trypomastigotes. For that matter, the authors fail to mention in the Introduction that metacyclogenesis happens at the final portion of the triatomine gut during the T. cruzi life cycle in the insect vector, and the known triggers of the process. As these are key to better understanding metacyclogenesis, these pieces of information should be added in the Introduction.

Page 2, Line 53

“…gene expression/genomic analyses

Results

Page 2, Figure 1 (same goes for Figure 2)

Replace ‘Malete’ for Malate

Replace ‘Piruvate’ for Pyruvate

Page 3, Lines 85 / 86

“Values correspond to the mean/average of organic acid concentration measured…”

Replace media for mean or average. It is not clear if the graph shows the standard deviation of the three biological replicates. If yes, add this information to the figure legend as it is in Figure 2. If not, add standard deviation to the graph. Also, there is no statistical analysis to show if the differences in the levels of the metabolites between the times points are significant. This should be added to strengthen the conclusions.

Page 3, Lines 95 to 97

Succinate was the only acid that increased in concentration (4-5 fold), but the whole time with low concentrations along with the culture

Confusing sentence. Consider rephrasing it.

Page 4, Lines 114 to 115

In the total extract from epimastigotes cultured in BHI media (Figure 4), the aldolase activity remained constant and appeared enhanced in the lag and log phases (24-72 h) as well as at the beginning of the stationary phase (120 h)

It is not clear what it is the standard of comparison for stating the aldolase activity ‘appeared enhanced’. Is it the activity of the other enzymes? Is it aldolase activity during metacyclogenesis? Please clarify in the text.

Page 4, Lines 115 to 119

On the other hand, the enzymatic activity of pyruvate kinase and hexokinase at 120 h of culture in BHI media was increased 4-5 fold and three-fold from 24 h, respectively. Comparing the profiles of enzyme activity in the extracts of parasites cultured in BHI media (Figure 4) to TAU3AAG (Fig. 3), the activity of three enzymes in the glycolytic pathway dropped during the first 24 h of differentiation.”

Again, there is no statistical analysis. From the data in the Supp. Tables, there is no significant difference in the pyruvate kinase activity in epimastigotes between 24h and 120h. The same goes for the hexokinase activity during the first 24h of differentiation.  Statistical analysis should be added and the conclusions rephrased.

Pages 4 and 5, Figures 3 and 4

To facilitate the comparison of the results, the y axis of the Pk and Hk activity graphs should be on the same scale.

Discussion

- The work of Barisón et al, 2016, used metabolomics to access the levels of 47 metabolites both in the exponential and stationary phases of T. cruzi epimastigotes. These analyses included the levels of citrate, pyruvate, malate and succinate. Barisón et al found a reduction in the levels of citrate, pyruvate, malate when parasites in the stationary phase were compared to parasites in the exponential phase. However, De-Simone and coauthors fail to compare and discuss these results. These should be added in the Discussion of the work along with possible explanations for divergent results.

- In the work of Adroher et al, 1990 the activities of T. cruzi’s hexokinase, phosphofructokinase and pyruvate kinase were measured in epimastigotes and metacyclic trypomastigotes. Although De-Simone and coauthors found similar results to the ones observed by Adroher and coauthors, this is not present in the Discussion and should be added.

Page 5, Line 134

in BHI medium (Figure 1) and its differentiation in TAU3AAG medium (Figure 2).”

The references for the figures are inverted in the Discussion. Please correct.

Also, as Figure 1 refers to the data obtained in epimastigotes and Figure 2 refers to the data obtained during metacyclogenesis, this reviewer believes that Figures 3 and 4 should follow the same pattern for consistency.

Page 6, Lines 150 and 151

“…type I reservosomes containing proteins and lipids disappear, and type II reservosomes containing…”

Please correct the typo.

Page 6, Lines 153 / 154

“The observation further supports this statement that the overall volume decreases during differentiation, and amino acid efflux is essential to parasite regulation.”

It is not clear what the authors mean and what it is the relevance of this information to the discussion of the work. Overall cell volume? Parasite regulation of what? Please clarify.

Page 6, Lines 159 to 161

Furthermore, a similar observation with the decreasing glycolytic enzymatic activity was obtained during amastigote differentiation to epimastigote forms [14, 37].”

These references are wrong. Reference 14 analyse the enzymatic activity of glycolytic enzymes during metacyclogenesis and should be discussed in detail, as mentioned above. Reference 37 refers to chromatography. The correct references should be added (like Engel et al, 1987) so the reviewers can access the strength of the conclusions.

Page 6, Paragraph starting on line 165

As this paragraph refers to the use of lipids during epimastigote and metacyclic trypomastigote stages, this should be moved to line 151 to improve clarity. Also, consider moving the reference to the end of the paragraph.

Page 6, Line 167

“…that the use of this fatty acid is irrelevant in this life stage.

Page 6, Line 170 to 172

The endogenous energy reserve present in various evolutive forms of T. cruzi has been attributed to triglycerides [47] and, while epimastigotes use glucose [46], the primary energy source in trypomastigotes are fatty acids [40].

Confusing statement concerning what is energy reserve and primary energy source. Consider rephrasing it. Also, reference 40 (Cazzulo, 1992) does not mention that the primary energy source in trypomastigotes is fatty acids. Please add the correct reference. Finally, many studies have shown the role of amino acids as energy sources for metacyclogenesis and metacyclic trypomastigotes, including Cazzulo, 1992. See also Contreras et al, 1985; Krassner et al, 1990 and Damasceno et al, 2018.

Material and Methods

Page 6, Lines 189 / 190

The Dm28c clone of Trypanosoma cruzi was maintained as previously described [52]. 189 harvested epimastigote forms after growth in BHI medium for 1, 3, 5, and 7 days.”

Confusing sentence. Consider rephrasing it.

Page 7, Topic 4.4

The method used for the preparation of protein extracts is not described. Also, the names of the kits used for the measurement of aldolase activity and protein quantification are not mentioned. These should be added.

Authors contributions

The contribution of the author GCL is not mentioned.

Supplementary Table 2

Replace 12h for 120h

Author Response

Referee 1

In the work entitled ‘Metabolic alteration of Trypanosoma cruzi during differentiation of epimastigote to trypomastigote forms’ De-Simone and co-authors use HPLC to quantify glycolytic and Krebs cycles short-chain carboxylic acids as well as to measure the enzymatic activities of aldolase, pyruvate kinase, and hexokinase in different time points of the life cycle of T. cruzi, the causing agent of Chagas disease. The authors found alterations in the levels of pyruvate, acetate, succinate, and glycerate when epimastigotes were compared to metacyclic trypomastigotes. Additionally, the authors show a strong reduction in the enzymatic activities of aldolase, pyruvate kinase, and hexokinase during the differentiation of epimastigote to the metacyclic trypomastigote form. Overall, the data is well presented. However, parts of the text need extensive English language and style editing to improve the clarity of the work. Moreover, Introduction, Material and Methods, Results, and Discussion sections must be improved.

R: Thank you for the comments and suggestions that will undoubtedly enhance our work.

Introduction

Page 1, Lines 31 /32: “…presents during its life cycle three main/major morphologically and physiologically distinct evolutive forms…”- Although the authors do not mention stages such as the intracellular epimastigotes (Almeida-de-Faria et al., 1999), adding the word ‘major’ or ‘main’ will implicate that there are other life stages of T. cruzi beyond the three mentioned stages.

R: Thank you, the word main was added; please see line 36.

Page 1, Lines 34 / 35-“…is its transformation into the highly infective metacyclic trypomastigote (trypo) forms.”-Although the authors mention the main morphological stages, it is important to differentiate metacyclic trypomastigotes from bloodstream trypomastigotes. For that matter, the authors fail to mention in the Introduction that metacyclogenesis happens at the final portion of the triatomine gut during the T. cruzi life cycle in the insect vector and the known triggers of the process. As these are key to better understanding metacyclogenesis, these pieces of information should be added in the Introduction.

R: New pieces of information were included as suggested.

Page 2, Line 53-“…gene expression/genomic analyses

R: Than you, done.

Results

Page 2, Figure 1 (same goes for Figure 2)-- Replace ‘Malete’ for Malate/-Replace ‘Piruvate’ for Pyruvate.

R: Thank you, do

Page 3, Lines 85 / 86-“Values correspond to the mean/average of organic acid concentration measured…”-Replace media for mean or average. It is not clear if the graph shows the standard deviation of the three biological replicates. If yes, add this information to the figure legend as it is in Figure 2. If not, add standard deviation to the graph. Also, there is no statistical analysis to show if the differences in the levels of the metabolites between the times points are significant. This should be added to strengthen the conclusions.

R: Thank you, the information has been added to the charts and discussion as suggested.

Page 3, Lines 95 to 97-“Succinate was the only acid that increased in concentration (4-5 fold), but the whole time with low concentrations along with the culture”-Confusing sentence. Consider rephrasing it.

R: Thank you, rephrased.

Page 4, Lines 114 to 115-“In the total extract from epimastigotes cultured in BHI media (Figure 4), the aldolase activity remained constant and appeared enhanced in the lag and log phases (24-72 h) as well as at the beginning of the stationary phase (120 h)”-It is not clear what it is the standard of comparison for stating the aldolase activity ‘appeared enhanced.’ Is it the activity of the other enzymes? Is it aldolase activity during metacyclogenesis? Please clarify in the text.

R: Yes, during the metacyclogenesis, we have modified the sentence.

Page 4, Lines 115 to 119- “On the other hand, the enzymatic activity of pyruvate kinase and hexokinase at 120 h of culture in BHI media was increased 4-5 fold and three-fold from 24 h, respectivelyComparing the profiles of enzyme activity in the extracts of parasites cultured in BHI media (Figure 4) to TAU3AAG (Fig. 3), the activity of three enzymes in the glycolytic pathway dropped during the first 24 h of differentiation.”

-Again, there is no statistical analysis from the data in the Supp. In tables, there is no significant difference in the pyruvate kinase activity in epimastigotes between 24h and 120h. The same goes for the hexokinase activity during the first 24h of differentiation. Statistical analysis should be added and the conclusions rephrased.

R: The statistical dates were included.

Pages 4 and 5, Figures 3 and 4- To facilitate the comparison of the results, the y axis of the Pk and Hk activity graphs should be on the same scale.

R: The figures were modified as suggested.

Discussion: - The work of Barisón et al., 2016, used metabolomics to access the levels of 47 metabolites both in the exponential and stationary phases of T. cruzi epimastigotes. These analyses included citrate, pyruvate, malate, and succinate levels. 

-Barisón et al. found a reduction in the levels of citrate, pyruvate, malate when parasites in the stationary phase were compared to parasites in the exponential phase. --However, De-Simone and coauthors fail to compare and discuss these results. These should be added in the Discussion of the work and possible explanations for divergent results.

R: Thank you, although the work evaluates the parasite growth metabolome and not the metacyclogenense one a sentence was included in the discussion. Please see, lines 200-204.

- In the work of Adroher et al., 1990 the activities of T. cruzi’ s hexokinase, phosphofructokinase, and pyruvate kinase were measured in epimastigotes and metacyclic trypomastigotes. Although De-Simone and coauthors found similar results to those observed by Adroher and coauthors, this is not present in the Discussion and should be added.

Page 5, Line 134-“in BHI medium (Figure 1) and its differentiation in TAU3AAG medium (Figure 2).”-The references for the figures are inverted in the Discussion. Please correct.

R: Thank you, corrected

As Figure 1 refers to the data obtained in epimastigotes and Figure 2 relates to the data obtained during metacyclogenesis, this reviewer believes that Figures 3 and 4 should follow the same pattern for consistency.

R: Thank you, the order of paragraphs was changed, as suggested.  

Page 6, Lines 150 and 151- “…type I reservosomes containing proteins and lipids disappear, and type II reservosomes containing…”- Please correct the typo.

R: The sentence typo was corrected.

Page 6, Lines 153 / 154-“The observation further supports this statement that the overall volume decreases during differentiation, and amino acid efflux is essential to parasite regulation.”- It is not clear what the authors mean and what it is the relevance of this information to the discussion of the work. Overall cell volume? Parasite regulation of what? Please clarify.

R: Thank you, was removed the information.

Page 6, Lines 159 to 161-“Furthermore, a similar observation with the decreasing glycolytic enzymatic activity was obtained during amastigote differentiation to epimastigote forms [14, 37].”- These references are wrong. Reference 14 analyzes the enzymatic activity of glycolytic enzymes during metacyclogenesis and should be discussed in detail, as mentioned above. Reference 37 refers to chromatography. The correct references should be added (like Engel et al., 1987) so the reviewers can access the strength of the conclusions.

R: Thank you, that is correct. Changed the references.

Page 6, Paragraph starting on line 165- This paragraph refers to the use of lipids during epimastigote, and metacyclic trypomastigote stages should be moved to line 151 to improve clarity. Also, consider moving the reference to the end of the paragraph.

R: Thank you made.

Page 6, Line 167-“…that the use of this fatty acid is irrelevant in this life stage.

R: Thank you, were introduced the words.

Page 6, Line 170 to 172- “The endogenous energy reserve present in various evolutive forms of T. cruzi has been attributed to triglycerides [47] and, while epimastigotes use glucose [46], the primary energy source in trypomastigotes are fatty acids [40].- Confusing statement concerning what is energy reserve and primary energy source. Consider rephrasing it. Also, reference 40 (Cazzulo, 1992) does not mention that the primary energy source in trypomastigotes is fatty acids. Please add the correct reference. Finally, many studies have shown the role of amino acids as energy sources for metacyclogenesis and metacyclic trypomastigotes, including Cazzulo, 1992. See also Contreras et al, 1985; Krassner et al, 1990 and Damasceno et al, 2018.

R: Thank you; was rephrased the sentence and the reference deleted.

Material and Methods

Page 6, Lines 189 / 190-“The Dm28c clone of Trypanosoma cruzi was maintained as previously described [52]. One hundred eighty-nine harvested epimastigote forms after growth in BHI medium for 1, 3, 5, and 7 days.”-Confusing sentence. Consider rephrasing it.

R: Thank was corrected.

Page 7, Topic 4.4-The method used for the preparation of protein extracts is not described. Also, the names of the kits used for the measurement of aldolase activity and protein quantification are not mentioned. These should be added.

R: The method is described in 4.2. Included the kit names.

Author's contributions- The contribution of the author GCL is not mentioned.

R: Thank you, do

Reviewer 2 Report

pathogens 1538742

The authors investigate a very interesting topic, metacyclogenesis of Trypanosoma cruzi.

A general suggestion is the replacement of Brazilian names of chemicals by English names, also in tables and figures (incl. Suppl. ones). In addition, the authors should change sentences with subordinate clauses beginning with “that”. In these sentences the first part can be deleted, sometimes by adding “In” or “According to” at the beginning. Thereby, the authors shorten the text and avoid repetitions of verbs such as report, show, demonstrate, suggest etc. (lines 38, 41, 50, 135, 138, 152, 158, 161, 165, 166, 167, 174, 176, 178, 181, 184). This is also relevant if an author is included in the text and not at the end of the sentence (line 147).

The following comments refer to the respective line.

17: The authors should replace “Here,” by “Here, we”

20: The authors should replace “B.H.I.” by the established abbreviation “BHI”, as done in line 65.

35: The authors should replace the Brazilian abbreviation “tripo” by “trypo”.

37: The authors should add: “Beside the transformations of epimastigotes also other ways of metayclogenesis occur in the vector. [Schaub, G.A.: Trypanosoma cruzi: quantitative studies of development of two strains in small intestine and rectum of the vector Triatoma infestans. Exp. Parasitol. 68, 260-273, 1989]

70, 71: The authors should replace “consists” by “consisted” and “reaches” by “reached” and throughout avoid changes between past and present tenses

76: The authors should delete one full stop.

76-80, 106-107: The authors should delete the sentences and include the indications at the end of the first sentence presenting the respective result.

82: The authors should replace “as a function of growth time” by “of parasites after different periods of growth”.

84: The authors should replace “at different values,” by “during cultivation”

85-87: The authors should present arithmetic means and standard deviations (or do they present medians?).

87: The authors should replace “varied from 5-30%” by “represent 5-30% of the total population”. Correct ?????

88: The authors should replace abbreviations if they are used <10 times.

89: The authors should replace “in” by “at” [throughout] and delete “studied”.

102: The authors should replace “realized with” by “determined in”.

104: The authors should replace “under” by “in”.

106, 113: The authors should describe the enzymatic activities in a sequence that is identical to that in the figures, i.e. thy should change the arrangement in the figures.

125: The authors should replace the subordinate clause by “and means an S.D. are presented”. ???

141: The authors should replace “bacterium” by “bacteria”.

151: The authors should replace “].” by “type”.

164: The authors should add: “However, this presumably does not reflect the natural changes in the rectum of the vector, the main site of metacyclogenesis, since there the presence of glucose is questionable.” (Guarneri, A.A.; Schaub, G.A.: Interaction of triatomines, trypanosomes and microbiota. In: Guarneri, A.A.; Lorenzo, M.G. (eds.) Triatominae – the biology of Chagas disease vectors. Springer Nature, New York, 345-386, 2021)

181: The authors should replace “that” by “whether or not monosaccharides are used by epimastigotes in the vector and that”.

189: The authors should replace “cloned” by “clone”.

191, 192, 198: The authors should add the centrifugation forces and the washing buffer.

193, 210, 216: Throughout, in all chemical formulae, the authors should type the numbers in a lower case not in a smaller size.

216: The authors should correct the chemical formula.

221: The authors should replace “measured changes” by “changes measured”.

221-222: The authors should delete the sentence up to [53].

230: The authors should replace “under anaerobic conditions” by “in a medium containing glucose”. [Or do the possess measurements of oxygen saturation??]

233: The authors should delete “extracellular” and should replace “epimastigote” by “of epimastigote”.

238-239: The authors should delete “Overlapping the total”.

240: The authors should replace “from” by “during”.

247: The authors should replace “in” by “at”.

240: The authors should replace “tripomastigote” by “trypomastigote”.

264: The authors should replace “G, DSci” by “G a PhD”. ????

The references contain too many mistakes. The authors should read and follow the Instructions more carefully. In the revised version, I will initially read the references and stop immediately, if mistakes are present. If the authors control separately for each of the following mistakes it is possible to submit references without mistakes. I stopped after no. 7 and found:

No.1: Volume not in italics.

Nos. 2, 4, 6: Add full stops after initials and semicolons between authors.

No. 3: Replace “Le T;” by “Le, T.”

No. 5: Delete full stops after initial abbreviations of journals.

No. 6: Replace short hyphen by a long dash in the page range.

No. 7: Do not abbreviate page numbering.

Author Response

Referee 2

The authors investigate a very interesting topic, metacyclogenesis of Trypanosoma cruzi.

A general suggestion is the replacement of Brazilian names of chemicals by English names, also in tables and figures (incl. Suppl. ones). In addition, the authors should change sentences with subordinate clauses beginning with “that”. In these sentences the first part can be deleted, sometimes by adding “In” or “According to” at the beginning. Thereby, the authors shorten the text and avoid repetitions of verbs such as report, show, demonstrate, suggest etc. (lines 38, 41, 50, 135, 138, 152, 158, 161, 165, 166, 167, 174, 176, 178, 181, 184). This is also relevant if an author is included in the text and not at the end of the sentence (line 147).

The following comments refer to the respective line.

17: The authors should replace “Here,” by “Here, we”

R: done

20: The authors should replace “B.H.I.” by the established abbreviation “BHI”, as done in line 65. R: Thank you, done

35: The authors should replace the Brazilian abbreviation “tripo” by “trypo”

R: Thank you, done.

37: The authors should add: “Beside the transformations of epimastigotes also other ways of metayclogenesis occur in the vector. [Schaub, G.A.: Trypanosoma cruzi: quantitative studies of development of two strains in small intestine and rectum of the vector Triatoma infestans. Exp. Parasitol. 68, 260-273, 1989] R: OK, done

70, 71: The authors should replace “consists” by “consisted” and “reaches” by “reached” and throughout avoid changes between past and present tenses

R:Than you

76: The authors should delete one full stop.

R: Thank you

76-80, 106-107: The authors should delete the sentences and include the indications at the end of the first sentence presenting the respective result.

82: The authors should replace “as a function of growth time” by “of parasites after different periods of growth”. (Fig 1 legends)

R:Done

84: The authors should replace “at different values,” by “during cultivation”

R: Done

85-87: The authors should present arithmetic means and standard deviations (or do they present medians?).

R: Thank you

87: The authors should replace “varied from 5-30%” by “represent 5-30% of the total population”. R: Correct, now line 97

88: The authors should replace abbreviations if they are used <10 times.

R: Thank you

89: The authors should replace “in” by “at” [throughout] and delete “studied”.

R: Thank you, now line 101.

102: The authors should replace “realized with” by “determined in”.]

R: OK, now line 114

104: The authors should replace “under” by “in”.

R: OK, done

106, 113: The authors should describe the enzymatic activities in a sequence that is identical to that in the figures, i.e. thy should change the arrangement in the figures.

R:

125: The authors should replace the subordinate clause by “and means an S.D. are presented”. R: Thank you, replaced

141: The authors should replace “bacterium” by “bacteria”. R: Thank you

151: The authors should replace “].” by “type”. R: Thank you, corrected

164: The authors should add: “However, this presumably does not reflect the natural changes in the rectum of the vector, the main site of metacyclogenesis, since there the presence of glucose is questionable.” (Guarneri, A.A.; Schaub, G.A.: Interaction of triatomines, trypanosomes and microbiota. In: Guarneri, A.A.; Lorenzo, M.G. (eds.) Triatominae – the biology of Chagas disease vectors. Springer Nature, New York, 345-386, 2021)

R: Thank you, done

181: The authors should replace “that” by “whether or not monosaccharides are used by epimastigotes in the vector and that”. R: Thank you

189: The authors should replace “cloned” by “clone”. R: Done

191, 192, 198: The authors should add the centrifugation forces and the washing buffer.

R: This information was present (1000 x g for 1 min), please see lines 228 to 235 in the new version.

193, 210, 216: Throughout, in all chemical formulae, the authors should type the numbers in a lower case not in a smaller size. R: Thank you was corrected

216: The authors should correct the chemical formula. R: Thank you was corrected

221: The authors should replace “measured changes” by “changes measured”. R: OK

221-222: The authors should delete the sentence up to [53].

R: OK

230: The authors should replace “under anaerobic conditions” by “in a medium containing glucose”. [Or do the possess measurements of oxygen saturation??]

R: OK

233: The authors should delete “extracellular” and should replace “epimastigote” by “of epimastigote”.

R: OK

238-239: The authors should delete “Overlapping the total”.

R: OK

240: The authors should replace “from” by “during”.

R: OK

247: The authors should replace “in” by “at”.

R: OK

240: The authors should replace “tripomastigote” by “trypomastigote”.

R: OK

264: The authors should replace “G, DSci” by “G a PhD”. ????

R: OK

The references contain too many mistakes. The authors should read and follow the Instructions more carefully. In the revised version, I will initially read the references and stop immediately, if mistakes are present. If the authors control separately for each of the following mistakes it is possible to submit references without mistakes. I stopped after no. 7 and found:

No.1: Volume not in italics. R: Ok

Nos. 2, 4, 6: Add full stops after initials and semicolons between authors.

R: OK

No. 3: Replace “Le T;” by “Le, T.”

R: OK

No. 5: Delete full stops after initial abbreviations of journals.

R: OK

No. 6: Replace short hyphen by a long dash in the page range.

No. 7: Do not abbreviate page numbering.

R: OK

Round 2

Reviewer 1 Report

After the first round of revisions, there were significant improvements in the clarity of the work. However, some of this reviewer’s comments have not been addressed:

First of all, revise the citation of the figures on the text and all the references throughout the text. There are many mistakes, some of them are detailed below. 

Introduction

Page 1, Lines 34 / 35 – Not addressed

“…is its transformation into the highly infective metacyclic trypomastigote (trypo) forms.”

Add metacyclic before trypomastigote. 

Results

Page 3, Lines 106 to 107 – Not addressed

Succinate was the only acid that increased in concentration (4-5 fold), but the whole time with low concentrations along with the culture

A confusing sentence that has not been rephrased.

Page 3 to 5 – Figure legends – New comment

It looks like the authors have done statistical analyses on the data shown in the graphs, but there is no mention of which test, method or software were used. This information should be added in the Figure legends as well as described in the Material and Methods.

Page 5, Lines 129 to 131 – Not addressed

In the total extract from epimastigotes cultured in BHI media (Figure 4), the aldolase activity remained constant and appeared enhanced in the lag and log phases (24-72 h) as well as at the beginning of the stationary phase (120 h)

It is not clear what it is the standard of comparison for stating the aldolase activity ‘appeared enhanced’. The authors clarified in the Author´s response that is during metacyclogenesis but the sentence has not been rephrased in the manuscript.

Page 5, Lines 131 to 133 – Not addressed

On the other hand, the enzymatic activity of pyruvate kinase and hexokinase at 120 h of culture in BHI media was increased 4-5 fold and three-fold from 24 h, respectively.

From the graph in Figure 4 and the data in the Supp. Table 4, there is no significant difference in the pyruvate kinase activity in epimastigotes between 24h, 72h and 120h. This sentence should be rephrased.

Page 5, Lines 133 to 136 – New comment

Comparing the profiles of enzyme activity in the extracts of parasites cultured in BHI media (Figure 4) to TAU3AAG (Fig. 3), the activity of three enzymes in the glycolytic pathway dropped during the first 24 h of differentiation.”

From the graphs in Figures 3 and 4 and the data in the Supp. Tables 3 and 4, there is no significant difference in the pyruvate kinase activity during the first 24 h of differentiation. This sentence should be rephrased.

Discussion

Not addressed:

- In the work of Adroher et al, 1990 the activities of T. cruzi’s hexokinase, phosphofructokinase and pyruvate kinase were measured in epimastigotes and metacyclic trypomastigotes. Although De-Simone and coauthors found similar results to the ones observed by Adroher and coauthors, this is not present in the Discussion and should be added.

Page 6, Line 151 – Not addressed

in BHI medium (Figure 1) and its differentiation in TAU3AAG medium (Figure 2).”

The references for the figures are still inverted in the Discussion. Please correct.

Also, as Figure 1 refers to the data obtained in epimastigotes and Figure 2 refers to the data obtained during metacyclogenesis, this reviewer believes that Figures 3 and 4 should follow the same pattern for consistency.

Page 6, Lines 172 to 174 – Not addressed

“The observation further supports this statement that the overall volumes decrease during differentiation, and amino acid efflux is essential to parasite regulation.”

It is not clear what the authors mean and what it is the relevance of this information to the discussion of the work. Overall cell volume? Parasite regulation of what? The authors responded that this sentence was removed from the text but is still there with no clarification.

Page 6, Lines 178 to 180 – Not addressed

Furthermore, a similar observation with the decreasing glycolytic enzymatic activity was obtained during amastigote differentiation to epimastigote forms [16,40].”

These references are still wrong. Reference 16 (Adroher et al, 1990) analyse the enzymatic activity of glycolytic enzymes during metacyclogenesis and should be discussed in detail, as mentioned above. Reference 40 (Toyota, 1995) refers to chromatography. The correct references should be added (like Engel et al, 1987).

Page 7, Line 192 – New comment

“…that the use of this fatty acid is irrelevant in this lifestyle.

Replace lifestyle for life stage.

Page 7, Lines 196 to 198 – Not addressed

The endogenous energy reserve present in various evolutive forms of T. cruzi has been attributed to triglycerides [55] and, while epimastigotes use glucose [56], the primary energy source in trypomastigotes are fatty acids [45,46].

Confusing statement concerning what is energy reserve and primary energy source. Consider rephrasing it. Also, references 45 (Cazzulo, 1992) and 46 (Urbina, 1994) do not mention that the primary energy source in trypomastigotes is fatty acids. Please add the correct references. Finally, many studies have shown the role of amino acids as energy sources for metacyclogenesis and metacyclic trypomastigotes, including Cazzulo, 1992 and Urbina, 1994. See also Contreras et al, 1985; Krassner et al, 1990 and Damasceno et al, 2018.

Page 7, Line 227

The Dm28c clone of Trypanosoma cruzi was maintained as previously described

Replace cloned by clone

Author Response

Introduction

 Page 1, Lines 34 / 35 – Not addressed  “…is its transformation into the highly infective metacyclic trypomastigote (trypo) forms.” Add metacyclic before trypomastigote.

R: OK, made 

 Results

Page 3, Lines 106 to 107 – Not addressed Succinate was the only acid that increased in concentration (4-5 fold), but the whole time with low concentrations along with the culture”. A confusing sentence that has not been rephrased.

R: Thank you, we have modified the sentence.

 Page 3 to 5 – Figure legends – New comment  It looks like the authors have done statistical analyses on the data shown in the graphs, but there is no mention of which test, method, or software were used. This information should be added in the Figure legends as well as described in the Material and Methods.

R: The information was inserted either in MM ( 4.5) and in the legends of all the figures.

Page 5, Lines 129 to 131 – Not addressed In the total extract from epimastigotes cultured in BHI media (Figure 4), the aldolase activity remained constant and appeared enhanced in the lag and log phases (24-72 h) as well as at the beginning of the stationary phase (120 h)”-   It is not clear what it is the standard of comparison for stating the aldolase activity 'appeared enhanced.' The authors clarified in the Author´s response that is during metacyclogenesis but the sentence has not been rephrased in the manuscript.

R: Thank you, we have modified the sentence.

 Page 5, Lines 131 to 133 – Not addressed On the other hand, the enzymatic activity of pyruvate kinase and hexokinase at 120 h of culture in BHI media was increased 4-5 fold and three-fold from 24 h, respectively.--From the graph in Figure 4 and the data in the Supp. Table 4, there is no significant difference in the pyruvate kinase activity in epimastigotes between 24h, 72h, and 120h. R: Thank you, this sentence was rephrased (lines 134-138).

Page 5, Lines 133 to 136 – New comment  “Comparing the profiles of enzyme activity in the extracts of parasites cultured in BHI media (Figure 4) to TAU3AAG (Fig. 3), the activity of three enzymes in the glycolytic pathway dropped during the first 24 h of differentiation.”--From the graphs in Figures 3 and 4 and the data in the Supp. Tables 3 and 4, there is no significant difference in the pyruvate kinase activity during the first 24 h of differentiation. This sentence should be rephrased.

R: Thank you, done (lines 134-138).

 Discussion

 Not addressed: - In the work of Adroher et al., 1990 the activities of T. cruzi’s hexokinase, phosphofructokinase and pyruvate kinase were measured in epimastigotes and metacyclic trypomastigotes. Although De-Simone and coauthors found similar results to the ones observed by Adroher and coauthors, this is not present in the Discussion and should be added.

R: Thank you, we introduced a paragraph in the discussion (please see lines 181-187).

 Page 6, Line 151 – Not addressed in BHI medium (Figure 1) and its differentiation in TAU3AAG medium (Figure 2).” -The references for the figures are still inverted in the Discussion. Please correct.

R: Thank you was modified.

Also, as Figure 1 refers to the data obtained in epimastigotes and Figure 2 refers to the data obtained during metacyclogenesis, this reviewer believes that Figures 3 and 4 should follow the same pattern for consistency.

R: The exact order of figures 1 and 2 were maintained in figures 3 and 4.

 Page 6, Lines 172 to 174 – Not addressed “The observation further supports this statement that the overall volumes decrease during differentiation, and amino acid efflux is essential to parasite regulation.”- It is not clear what the authors mean and what it is the relevance of this information to the discussion of the work. Overall cell volume? Parasite regulation of what? The authors responded that this sentence was removed from the text but is still there with no clarification.

R: Thank you, we removed this sentence to avoid confusion.

Page 6, Lines 178 to 180 – Not addressed Furthermore, a similar observation with the decreasing glycolytic enzymatic activity was obtained during amastigote differentiation to epimastigote forms [16,40].”- These references are still wrong. Reference 16 (Adroher et al., 1990) analyses the enzymatic activity of glycolytic enzymes during metacyclogenesis and should be discussed in detail, as mentioned above. Reference 40 (Toyota, 1995) refers to chromatography. The correct references should be added (like Engel et al., 1987).

R: Correct, reference 40 was removed, and reference Engel et al.[53] introduced.

 Page 7, Line 192 – New comment “…that the use of this fatty acid is irrelevant in this lifestyle.Replace lifestyle for life stage.

R: OK, modified.

Page 7, Lines 196 to 198 – Not addressed The endogenous energy reserve present in various evolutive forms of T. cruzi has been attributed to carbohydrate [55]. However, while epimastigotes can interchange glucose and amino acids, it also can accumulates fatty acids into lipid droplets during growth and use this reserve to support its growth after glucose exhaustion [Souza et al., 2021; 56]. it as the primary energy source in metacyclic trypomastigotes are carbohydrates (mainly glucose) and amino acids acids (mostly proline) [XXXXX].-Confusing statement concerning what is energy reserve and primary energy source. Consider rephrasing it.

Also, references 45 (Cazzulo, 1992) and 46 (Urbina, 1994) do not mention that the primary energy source in trypomastigotes is fatty acids. Please add the correct references.

R: Thank you, the paragraph was modified and new references were introduced (lines 198-203)

Finally, many studies have shown the role of amino acids as energy sources for metacyclogenesis and metacyclic trypomastigotes, including Cazzulo, 1992 and Urbina, 1994. See also Contreras et al, 1985; Krassner et al, 1990 and Damasceno et al, 2018.

R: OK, these references [3, 5, 24] were cited (see 219).

 Page 7, Line 227 “The Dm28c clone of Trypanosoma cruzi was maintained as previously described”- Replace cloned by clone.

R: Ok, made
